# Molecular Identification of *Blastocystis hominis* Isolates in Patients with Autoimmune Diseases

Ahmed M. Mahmoud [1,*,†], Khadiga A. Ismail [2,†], Osama M. Khalifa [3], Maha M. Abdel-wahab [4], Howaida M. Hagag [2] and Mahmoud K. Mahmoud [5]

1   Department of Urology, Mayo Clinic, Rochester, MN 55905, USA
2   Department of Clinical Laboratory Sciences, College of Applied Medical Sciences, Taif University, Taif 21994, Saudi Arabia
3   Department of Internal Medicine, Faculty of Medicine, Ain Shams University, Cairo 11566, Egypt
4   Department of Medical Parasitology, King Faisal Medical Complex, Taif 26724, Saudi Arabia
5   Department of Dermatology, King Faisal Medical Complex, Taif 26724, Saudi Arabia
*   Correspondence: mahmoud.ahmed@mayo.edu; Tel.: +1-(507)-606-9066
†   These authors contributed equally to this work.

**Abstract:** Background: *Blastocystis hominis* (*B. hominis*) is a ubiquitous parasite that has spread worldwide and is commonly present in human stool specimens. It was hypothesized that infection with *B. hominis* plays a role in the pathogenesis of autoimmune diseases in humans. The aim of this study is to test this hypothesis by investigating patients with autoimmune diseases. Patients with various types of autoimmune diseases with gastrointestinal symptoms were enrolled in this study as cases (*n* = 72) along with nongastrointestinal symptom patients as controls (*n* = 58). All participants in this study were subjected to history taking and were investigated for *B. hominis* infection via wet-mount microscopic stool examinations, staining with trichrome stain, and molecular-based tests applied to their fecal samples. Blood samples were also tested for complete blood counts. *B. hominis* were identified with specific PCR more in cases (12/72; 16.6%) than in controls (3/58; 5.2%), with a significant difference ($p < 0.05$). Significant decreases in white blood cell counts were demonstrated in systemic lupus erythematosus (SLE) and ulcerative colitis (UC) patients infected with *B. hominis* when compared to patients with nongastrointestinal symptoms (*p*-value $< 0.05$).

**Keywords:** *Blastocystis hominis*; autoimmune diseases; molecular diagnosis

## 1. Introduction

The parasite *Blastocystis hominis* (*B. hominis*), which was first described by a Russian scientist in 1870, was initially ignored because it had no assigned taxonomic rank. In 1912, it was identified in stool samples as a harmless yeast, although it went unnoticed for many years. In comparison with other intestinal parasites, it is now understood to be the most prevalent and is regarded as a strange parasite. There are a growing number of reports of *B. hominis* being isolated from human fecal material, particularly in tropical nations, and theories about its pathogenicity have brought more attention to it [1].

*Blastocystis hominis* (*B. hominis*) is an anaerobic, eukaryotic, unicellular protozoan with a global distribution that lives in the human large intestine [2]. *B. hominis* has colonized between one and two billion people worldwide [3,4]. Evidence of *B. hominis* detection in the feces of mammals and birds points to the parasite's zoonotic nature and suggests that these animals may be the cause of many human infections [5]. *B. hominis* has been classified into four main morphological forms: vacuolar, granular, ameboid, and cystic [3,6]. The fecal–oral route is the main way that *B. hominis* infections spread [4].

In humans and a variety of animals, 22 subtypes of *Blastocystis* have been discovered based on investigations of the small subunit (SSU) rRNA gene. These subtypes may represent different species [7]. The prevalence of *B. hominis* infections was found in previous

studies to be around 50% in developing countries [8,9] and reached up to 20% in developed countries [10,11]. The differences in prevalence were caused by factors that are more prevalent in developing countries, such as poor sanitary conditions, poor personal hygiene, close contact with animals, and consumption of infected food or water [12].

Evidently, the issue of *B. hominis'* pathogenic potential for human health was brought up by the high prevalence of infected cases in the species that were reported [13,14]. Unspecific gastrointestinal symptoms, such as diarrhea, abdominal pain, vomiting, irritable bowel syndrome (IBS) [15], and urticaria, have been linked to the parasite in prior research [16]. On the basis of different SSU-rRNA gene sequences, other researchers have linked the aforementioned symptoms to specific clinical isolates [17]. Laboratory mouse models infected with the parasite have been shown to suffer from oxidative damage [18], and the intestinal permeability of patients with *Blastocystis* infections was also significantly higher than that of healthy individuals [19].

Autoimmune diseases can occur as a result of an abnormal immune response to a normal body part due to an unexplained etiology. Systemic lupus erythematosus (SLE) is one example of an autoimmune disorder that has a family history. Other autoimmune diseases include multiple sclerosis, psoriasis, rheumatoid arthritis (RA), Graves' disease, ulcerative colitis (UC), diabetes mellitus type 1, and Crohn's disease (CD) [20]. In approximately 3% of people worldwide, autoimmune disorders are prevalent. Compared to men, women were reported to have a higher prevalence rate. Individuals with insufficient or nonfunctional immunoregulatory mechanisms toward environmental pathogens are prone to autoimmune disorders [21–23].

Systemic lupus erythematosus (SLE) is characterized by the development of antibodies against several cell nucleus components and exhibits a wide range of clinical symptoms. Inflammation, vasculitis, immune complex deposition, and vasculopathy are the main pathology signs in autoimmune disease patients with SLE. To confirm the precise etiology of SLE, there are numerous theories. With a significantly higher prevalence among patients' first-degree relatives, SLE exhibits a substantial familial aggregation. Additionally, SLE and other organ-specific autoimmune disorders, such as thyroiditis, immune thrombocytopenic purpura, and hemolytic anemia, may coexist in extended families. Despite the fact that most occurrences of SLE are sporadic and lack genetic predisposing factors, this suggests that other environmental or as-of-yet unidentified variables may also be to blame [24]. Keshawy and Alabbassy [25] showed that Blastocystis hominis infestations, mainly subtypes 2 and 3, were found in 41.66% of the studied SLE patients, which is significantly higher than the percentage found in either IBS patients or healthy controls. This suggests a possible link between Blastocystis hominis and the SLE disease process.

Rheumatoid arthritis (RA), a systemic inflammatory autoimmune disease that affects the articular surfaces of the joints, progresses over time. Although the precise etiology is uncertain, both genetic and environmental factors play a role. In the pathophysiology of RA, T cells, B cells, and the controlled interplay of proinflammatory cytokines play important roles [26]. In patients with RA, intestinal protozoa infections are linked to increased intestinal permeability. Intestinal protozoa should, therefore, be clinically examined because they may contribute to the clinical heterogeneity of the disease. Additionally, *Blastocystis sp.* infections are common in southern Mexican RA patients [27].

One of the significant inflammatory bowel diseases, ulcerative colitis (UC), is still poorly understood in terms of its pathophysiology. The inflammation and progression of the disease are thought to be mediated by several hereditary variables and other genes associated with inflammation. It has been discovered that several susceptibility loci linked to a higher risk of ulcerative colitis are related to mucosal barrier function. Environmental factors, including smoking, oral contraceptives, food, antibiotics, vaccinations, infections, and hygiene, as well as several biomarkers that harm the intestinal mucosa, all play a part [28]. In addition to producing cysteine proteases that are nuclear-factor-B-dependent, *B. hominis* also induces the expression of IL-6, IL-1, and TNF-α, which upregulate proinflammatory cytokines, and activates mitogen-activated protein kinase (MAPK) in macrophages,

all of which support the role of serine proteases in *Blastocystis* virulence. Patients with IBD are more likely to develop a *Blastocystis* spp. infection, particularly those with UC. Patients with symptomatic Crohn's disease (CD) are also more likely to contract this infection [5].

A hereditary susceptibility, autoimmune pathogenic inflammation, uncontrolled keratinocyte proliferation, and defective differentiation are all features of the chronic inflammatory skin condition psoriasis. The inflammatory infiltrates of the psoriatic plaques are formed of dermal dendritic cells, macrophages, T cells, and neutrophils, and are overlaid with acanthosis (epidermal hyperplasia), according to the histology of the condition. Neovascularization is another noticeable characteristic. Inflammation has been demonstrated to impact several organ systems in addition to the psoriatic skin. Due to vulnerability, psoriasis has been linked to a higher occurrence of gastrointestinal conditions. This relationship is supported, particularly with regard to Crohn's disease, by loci shared between psoriasis and inflammatory bowel disease [29].

Eczema (E) is a chronically recurrent, itchy, and inflamed skin condition. Increased Th2 activity is shown in E, both in the skin and the bloodstream. It is based in part on a person's genetic makeup (for example, mutations in the IgE receptor), as well as the type of antigen that triggers an immunological response. Plasma cells produce IgE antibodies as a result of Th2 cell stimulation. Defective cellular immunity has been hypothesized because people with E are more likely to develop a wide range of infectious disorders, including fungal, viral, bacterial, and staphylococcal impetigo [30].

There have been reports indicating that irritable bowel syndrome (IBS) is brought on by an inflammatory disease of the intestine that may be associated with *Blastocystis* infections [31]. In order to cause inflammation, it breaks down the epithelial barrier and raises the levels of proinflammatory cytokines [32]. The intestinal barrier collapses, which results in oxidative damage, triggering an innate immune response and leading to the symptoms of a *Blastocystis* infection. The intestinal epithelium is invaded and damaged, and this activation of TLRs and CD8 T cells, macrophages, and neutrophils results in the formation of immunoglobulin M (IgM), IgG, and IgA [33]. However, it is still unclear how *Blastocystis* colonization affects gastrointestinal symptoms.

The following are some of *B. hominis*' virulence factors: cysteine proteases, cyclophilin-like proteins, serine proteases, aspartic proteases, sugar-binding proteins, metalloproteases, glycosyltransferases, hydrolases of the glucoside–hydrolase subclass, and protease inhibitors. These elements are examples of the proteases that are present and are thought to contribute to disease development at the pathogen–host interface, and the immune response involves a reaction to digesting enzymes or protease I. The presence of proteins with immunoglobulin-like domains in the *Blastocystis* genome may also point to mediators of adhesion to host cells. Type 1 polyketide synthase (PKS), nonribosomal peptide synthase (NRPS), and superoxide dismutase (SOD) with iron are all present in *Blastocystis*. The ability of *Blastocystis* to generate metabolites, such as simple fatty acids, and many chemicals, including poisons, all point to the pathogenic potential of this organism [34].

The intestinal lumen's IgA secretion serves as a barrier against invasive germs [35]. Patients with *Blastocystis* infections have lower fecal IgA levels than people who are not colonized [36]. Additionally, there is a link between *Blastocystis* and lower blood neutrophil numbers [37], and it has been known for producing serine proteases that break down secretory IgA (sIgA) [38]. It is believed that *B. hominis* has the ability to alter the tight junctions between the intestinal epithelial cells and the intestinal content in order to impair barrier function and contribute to the pathogenesis of autoimmune diseases in people [39]. Patients with autoimmune diseases were observed in the current study to investigate this theory. Patients with various autoimmune disease illness spectra and who had gastrointestinal manifestations were included in this study as cases, whereas patients with autoimmune diseases but without gastrointestinal manifestations were included as controls. Medical histories were taken from patients and controls, and fecal samples were tested using microscopy and molecular methods to look for *B. hominis* infections.

## 2. Materials and Methods

### 2.1. Patients and Settings

A case–control study was carried out between January 2022 and January 2023. Inclusion criteria were patients with autoimmune diseases under different protocols of treatment according to their respective illness. Patients who had recently used antibiotics and antiprotozoals and patients with comorbid chronic illnesses other than autoimmune disorders were excluded, including those with diabetes and/or chronic liver and/or kidney conditions. In our study, 130 autoimmune disorders were divided into the following groups: 72 individuals with gastrointestinal problems (cases) and 58 autoimmune disease patients with nongastrointestinal symptoms (including 38 SLE, 36 RA, and 21 UC). Thirteen Crohn's, thirteen psoriasis, and nine eczema patients were recruited for the study in addition to the cases. Age and sex matching were taken into consideration when choosing the cases and controls.

### 2.2. Data and Sample Collection

Thorough histories were taken for all cases and controls in order to collect demographic and clinical information. Each participant was required to deliver one feces sample each day for three days. Each stool sample was separated into three aliquots, one of which was maintained fresh without a preservative, one in 10% formalin, and the third in pure alcohol. Each participant was asked to contribute 5 mL of venous blood in addition to the fecal samples. A full blood count test was performed on the blood. On the day of collection, samples were delivered to the clinical laboratory where they were carefully processed before being used for the studies.

### 2.3. Laboratory Investigations

Feces were examined microscopically to look for trophozoites, protozoan cysts, or parasite eggs. To identify different parasite stages, wet mounts of fresh and preserved feces were stained with saline and iodine. A trichrome-stained fecal smear was also used for parasite detection. Both fresh and formol-ether-concentrated feces were used for the parasitological examination.

### 2.4. Blastocystis Hominis Stool PCR

The alcohol-kept fecal aliquots were treated to DNA extraction using a QIA amp® fast DNA stool mini kit for genomic DNA purification (cat. No. 51604 Qiagen, Hilden, Germany), following the kit's instructions. In accordance with a test that had already been reported, DNA extracts were submitted to PCR amplification [40]. In a LightCycler, the reaction setup and heat cycles were carried out. (Roche Diagnostics Corporation, 100 Mannheim, Germany). GoTaq Hot Start Polymerase (Promega, Madison, WI, USA) and other PCR reagents employed in the amplification procedures had final concentrations that were very similar to the assay that had previously been reported. Briefly, adhering to the manufacturer's instructions and using a previously described primer pair, a 600 bp DNA sequence from a highly conserved region of the small subunit rRNA gene of *Blastocystis hominis* was amplified. Forward primer was (5′-GGA GGT AGT GAC AAT AAA TC-3′) and reverse primer was (5′-TGC TTT CGC ACT TGT TCA TC-3′). Amounts of 50 μL of Go Tag® Green master mix (Promega, Madison, WI, USA), 10 μL of DNA extract, and 0.8 μg/μL of each primer were used in the reaction setup. The target DNA sequence was amplified using 40 cycles of denaturation at 90 °C for 1 min, annealing at 56 °C for 2 min, and primer extension at 72 °C for 1 min, followed by further extension at 72 °C for 5 min. External controls that were both positive and negative were added to the test samples in each PCR cycle. The negative control was a DNA-free blank with all of the PCR reagents, whereas the positive control had DNA extracted from the pooled stool samples that tested positive for *Blastocystis hominis*. For the purpose of PCR amplification detection, about 10 μL of each amplification reaction and the DNA molecular size marker were electrophoresed in a 1.5%

agarose gel for 1 h, stained with a 0.5 μg/mL ethidium bromide, and, lastly, visualized using ultraviolet transilluminator (BIO-RAD Lab., Hercules, California, USA).

### 2.5. Data Processing and Statistical Analysis

The results were examined using SPSS version 22 (IBM Inc., Armonk, NY, USA) computer software. Unpaired t-tests, exact Fisher tests, and chi-square tests were all applied. A *p*-value of less than 0.05 was regarded as statistically significant for all two-sided tests.

### 2.6. Ethical Considerations

This study was approved by Taif University's and KFMC ethical council as being ethical.(H-02-T-123) Prior to their involvement, all patients who were recruited for our study were informed of its goals and its methods.

### 3. Results

This study was performed on 72 cases (22 males and 50 females) with a mean age of $37.53 \pm 13.4$ and 58 controls (18 males and 40 females) with a mean age of $40.9 \pm 12.4$.

Table 1 outlines the common patient demographics of both cases and controls.

**Table 1.** Distribution of patients regarding demographics and different autoimmune diseases in *Blastocystis*-positive patients and *Blastocystis*-negative patients with PCR.

| Variable | | Blastocystis sp. Infection (*n*, %) | | OR | 95% CI for OR | *p* |
|---|---|---|---|---|---|---|
| | | Positive (*n* = 15) | Negative (*n* = 115) | | | |
| Age | <40 (*n* = 82) | 11 (73.3%) | 71 (61.7%) | 1.70 | 0.55–5.10 | 0.38 |
| | ≥40 (*n* = 48) | 4 (26.7%) | 44 (38.3%) | | | |
| Gender | Male (*n* = 40) | 9 (60.0%) | 51 (44.3%) | 1.88 | 0.64–5.76 | 0.25 |
| | Female (*n* = 90) | 6 (40.0%) | 64 (55.7%) | | | |
| Residence | Urban (*n* = 116) | 9 (60.0%) | 107 (93.0%) | 0.11 | 0.03–0.40 | <0.01 * |
| | Rural (*n* = 14) | 6 (40.0%) | 8 (7.0%) | | | |
| SLE | No (*n* = 92) | 10 (66.7%) | 82 (71.3%) | 0.80 | 0.27–2.25 | 0.71 |
| | Yes (*n* = 38) | 5 (33.3%) | 33 (28.7%) | | | |
| RA | No (*n* = 94) | 14 (93.3%) | 80 (69.6%) | 6.13 | 1.07–66.72 | 0.06 |
| | Yes (*n* = 36) | 1 (6.7%) | 35 (30.4%) | | | |
| UC | No (*n* = 109) | 10 (66.7%) | 99 (86.1%) | 0.32 | 0.09–0.97 | 0.06 |
| | Yes (*n* = 21) | 5 (33.3%) | 16 (13.9%) | | | |
| Crohn's | No (*n* = 117) | 13 (86.7%) | 104 (90.4%) | 0.69 | 0.15–3.39 | 0.65 |
| | Yes (*n* = 13) | 2 (13.3%) | 11 (9.6%) | | | |
| Psoriasis | No (*n* = 117) | 14 (93.3%) | 103 (89.6%) | 1.63 | 0.24–18.64 | 0.65 |
| | Yes (*n* = 13) | 1 (6.7%) | 12 (10.4%) | | | |
| Eczema | No (*n* = 121) | 14 (93.3%) | 107 (93.0%) | 1.05 | 0.17–12.38 | 0.97 |
| | Yes (*n* = 9) | 1 (6.7%) | 8 (7.0%) | | | |

Abbreviations: OR: odd ratio; CI: confidence interval; SLE: systemic lupus erythematosus; RA: rheumatoid arthritis; UC: ulcerative colitis; * Statistically significant.

Table 2 and Figure 1 show hematological parameters in cases and controls of different autoimmune disease patients.

**Table 2.** Hematological parameters in cases and controls of patients with different autoimmune diseases.

| Diseases | | Case (Mean, ±SD) | Control (Mean, ±SD) | *p* |
|---|---|---|---|---|
| SLE | WBCs | 7.39, (2.65) | 9.12, (2.41) | 0.04 * |
| | Neutrophiles | 4.76, (2.19) | 5.79, (1.99) | 0.14 |
| | Lymphocytes | 2.35, (1.19) | 2.22, (0.97) | 0.7 |
| | Monocytes | 0.94, (0.53) | 0.99, (0.49) | 0.75 |
| | Eosinophiles | 0.13, (0.15) | 0.11, (0.14) | 0.8 |
| RA | WBCs | 8.05, (2.51) | 7.52, (2.84) | 0.56 |
| | Neutrophiles | 5.09, (1.86) | 4.49, (2.23) | 0.39 |
| | Lymphocytes | 2.46, (0.88) | 2.67, (0.95) | 0.52 |
| | Monocytes | 0.91, (0.49) | 0.78, (0.52) | 0.44 |
| | Eosinophiles | 0.10, (0.15) | 0.11, (0.11) | 0.79 |
| UC | WBCs | 9.06, (3.58) | 9.51, (4.62) | 0.8 |
| | Neutrophiles | 5.07, (2.55) | 18.85, (19.93) | 0.03 * |
| | Lymphocytes | 3.31, (1.32) | 16.59, (18.89) | 0.02 * |
| | Monocytes | 1.01, (0.54) | 4.23, (5.42) | 0.053 |
| | Eosinophiles | 0.22, (0.19) | 0.71, (0.98) | 0.11 |
| Crohn's | WBCs | 7.04, (2.17) | 6.83, (2.64) | 0.88 |
| | Neutrophiles | 3.76, (1.47) | 10.61, (15.09) | 0.22 |
| | Lymphocytes | 2.52, (1,13) | 11.58, (21.14) | 0.24 |
| | Monocytes | 0.62, (0.16) | 2.26, (4.15) | 0.28 |
| | Eosinophiles | 0.14, (0.09) | 0.68, (1.24) | 0.23 |
| Psoriasis | WBCs | 8.66, (1.92) | 9.39, (5.12) | 0.73 |
| | Neutrophiles | 5.47, (1.59) | 5.54, (4.08) | 0.97 |
| | Lymphocytes | 2.31, (1.19) | 3.51, (1.70) | 0.16 |
| | Monocytes | 0.84, (0.48) | 0.69, (0.43) | 0.55 |
| | Eosinophiles | 0.12, (0.15) | 0.18, (0.19) | 0.54 |
| Eczema | WBCs | 5.51, (1.31) | 8.40, (3.94) | 0.21 |
| | Neutrophiles | 3.26, (1.26) | 4.06, (2.42) | 0.57 |
| | Lymphocytes | 1.79, (0.48) | 3.61, (1.75) | 0.09 |
| | Monocytes | 0.71, (0.36) | 0.69, (0.29) | 0.98 |
| | Eosinophiles | 0.07, (0.09) | 0.34, (0.22) | 0.06 |

Abbreviations: SLE: systemic lupus erythematosus; RA: rheumatoid arthritis; UC: ulcerative colitis; * Statistically significant.

In Table 3 the parasite is identified, with a significant difference in the levels of positivity of *Blastocystis* infection between cases and controls ($p < 0.05$) who had SLE and UC.

Tables 4 and 5 show the *B. hominis* positivity variations between direct microscopy (all the detected isolates were of the vacuolar type) and specific PCR assay. The overall detection rate of *Blastocystis* was 7.6% (10 out of 130) with direct microscopy; however, the rate was 11.5% (15 out of 130) with specific PCR assay.

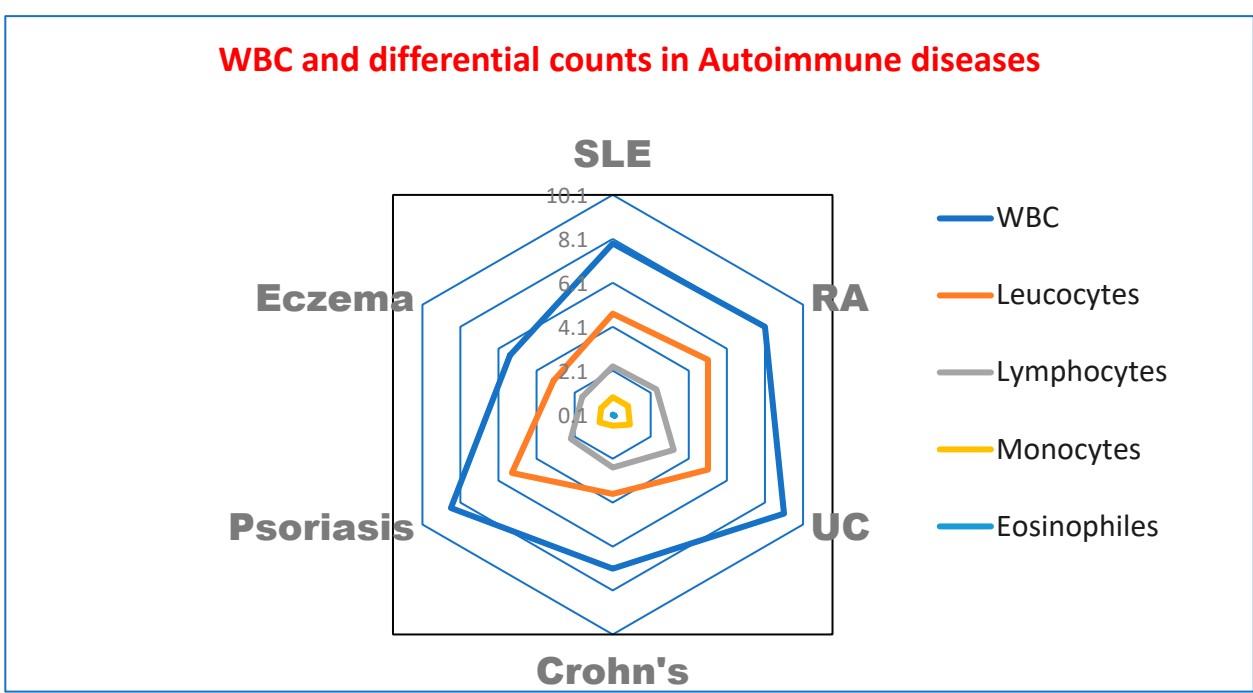

**Figure 1.** Radar plot of hematological parameters in cases and controls of patients with different autoimmune diseases. WBC: white blood cells.

**Table 3.** Distribution of *Blastocystis*-positive patients among different autoimmune diseases (cases and controls) with PCR.

| Diseases | *Blastocyst* | Case (*n*) | Control (*n*) | *p* |
|---|---|---|---|---|
| SLE | Positive (*n* = 5) | 5 | 0 | 0.02 * |
| | Negative (*n* = 33) | 15 | 18 | |
| RA | Positive (*n* = 1) | 0 | 1 | 0.43 |
| | Negative (*n* = 35) | 21 | 14 | |
| UC | Positive (*n* = 5) | 5 | 0 | 0.04 * |
| | Negative (*n* = 16) | 7 | 9 | |
| Crohn's | Positive (*n* = 2) | 2 | 0 | 0.49 |
| | Negative (*n* = 11) | 6 | 5 | |
| Psoriasis | Positive (*n* = 1) | 0 | 1 | 0.99 |
| | Negative (*n* = 12) | 7 | 5 | |
| Eczema | Positive (*n* = 1) | 0 | 1 | 0.99 |
| | Negative (*n* = 8) | 4 | 4 | |

Abbreviations: SLE: systemic lupus erythematosus; RA: rheumatoid arthritis; UC: ulcerative colitis; * Statistically significant.

**Table 4.** Diagnostic test results of 130 studied patients.

| Test | Total 130 | | Cases | | Control | | Test of Sign | |
|---|---|---|---|---|---|---|---|---|
| | Positive *n* | Negative *n* | Positive *n* | Negative *n* | Positive *n* | Negative *n* | $\chi^2$ | *p* Value |
| Microscopy | 10 | 120 | 9 | 63 | 1 | 57 | 5.245 | 0.02 * |
| PCR | 15 | 115 | 12 | 60 | 3 | 54 | 4.03 | 0.04 * |

* Statistically significant.

**Table 5.** Diagnostic performance of microscopy versus the PCR test's findings.

| Test | Positive | | Negative | | Sensitivity % (95% CI) | Specificity % (95% CI) | PPV % (95% CI) | NPV % (95% CI) | Agreements % (Kappa Test) |
|---|---|---|---|---|---|---|---|---|---|
| | True | False | True | False | | | | | |
| Wet mount microscopy | 10 | 0 | 120 | 5 | 66.67% (38.38–88.18) | 100 (96.97–100.0) | 100 | 96.77% (92.9–98.56) | 80.1 (0.353) |
| PCR assay | 15 | 0 | 115 | 0 | NA | NA | NA | NA | NA |

Abbreviations: CI: confidence intervals; PPV: positive predictive value; NPV: negative predictive value; NA: not applicable.

## 4. Discussion

According to reports, *B. hominis* can suppress the host's immune response and has a harmful function in autoimmune illnesses [41]. Growing evidence points to a potential connection between the gut microbiota and the development of UC, either directly by producing inflammation or indirectly through a compromised immune system [42]. Despite *Blastocystis* being a typical resident of the human digestive tract, further research is still needed to determine how it contributes to UC. As the gold-standard test, specific PCR was used in the current investigation to determine the prevalence rate of *B. hominis* in cases and controls. Contrary to earlier reports [43], the rate of *Blastocystis* infection was not gender-related.

According to Table 1 of the current study, patients who live in rural regions had significantly higher prevalence rates of *B. hominis*, which is consistent with other studies that have found several risk factors linked to *Blastocystis* sp. Poor hygiene practices, drinking nontap water, not washing hands after using the bathroom, and contact with animals are all examples of infection risk factors [44]. Additionally, there were correlations between the parasite's prevalence and low socioeconomic status, low education, and poor health [45]. Furthermore, UC patients were found to be at very high risk of *B. hominis* infection, which is in agreement with Mumcuoglu et al. [46]. They previously reported that *Blastocystis* was more common among IBS patients and that symptoms subsided following *Blastocystis* treatment. In a different study, 99 individuals who tested positive for *Blastocystis* were compared to a control group, and it was discovered that none of the gastrointestinal symptoms were caused by *Blastocystis* infection [47]. *Blastocystis* should be investigated in UC patients when the symptoms are resistant, regardless of these debates [48].

In accordance with the study conducted by Cheng et al., which found that hemoglobin, neutrophil count, and hematocrit were decreased in subjects with *B. hominis* infections, as *Blastocystis hominis* is a possible factor in hematological abnormalities [37], Table 2 shows a significant decrease in hematological parameters in cases of SLE and UC compared to controls of autoimmune diseases patients. In line with Yakoob et al. [49], who observed a greater incidence of *Blastocystis* sp. in an IBS patient population in Pakistan, Table 3 demonstrates significant increases in *B. hominis* infections in UC patients. However, a study conducted by French researchers, Nourrisson et al. [50], did not reveal any prevalence. The variation in the number of IBS patients included in the respective trials and the differences in the geographical locations of the two studies may perhaps account for the differences in *Blastocystis* sp. prevalence in each clinical subgroup of IBS.

When compared to the wet mount approach, the PCR showed the best results. It has repeatedly been noted that the microscope has low sensitivity for detecting infections. The sensitivity of common microscopy varied between 38% and 82%, according to one study [51]. Comparative analysis showed that the sensitivity of the microscopic detection for *Blastocystis* was 73.4%, whereas it was 90.6% for in vitro cultures and 95.9–96.7% when using the PCR detection method. Their results were marginally higher than the findings of this study, which showed that the sensitivity of microscopic detection for *Blastocystis* was 66.67%, as shown in Table 4, Table 5 and Figure 2. When comparing several techniques, Roberts et al., Eida, and Eida [52,53] discovered that PCR was most effective at finding *Blastocystis*.

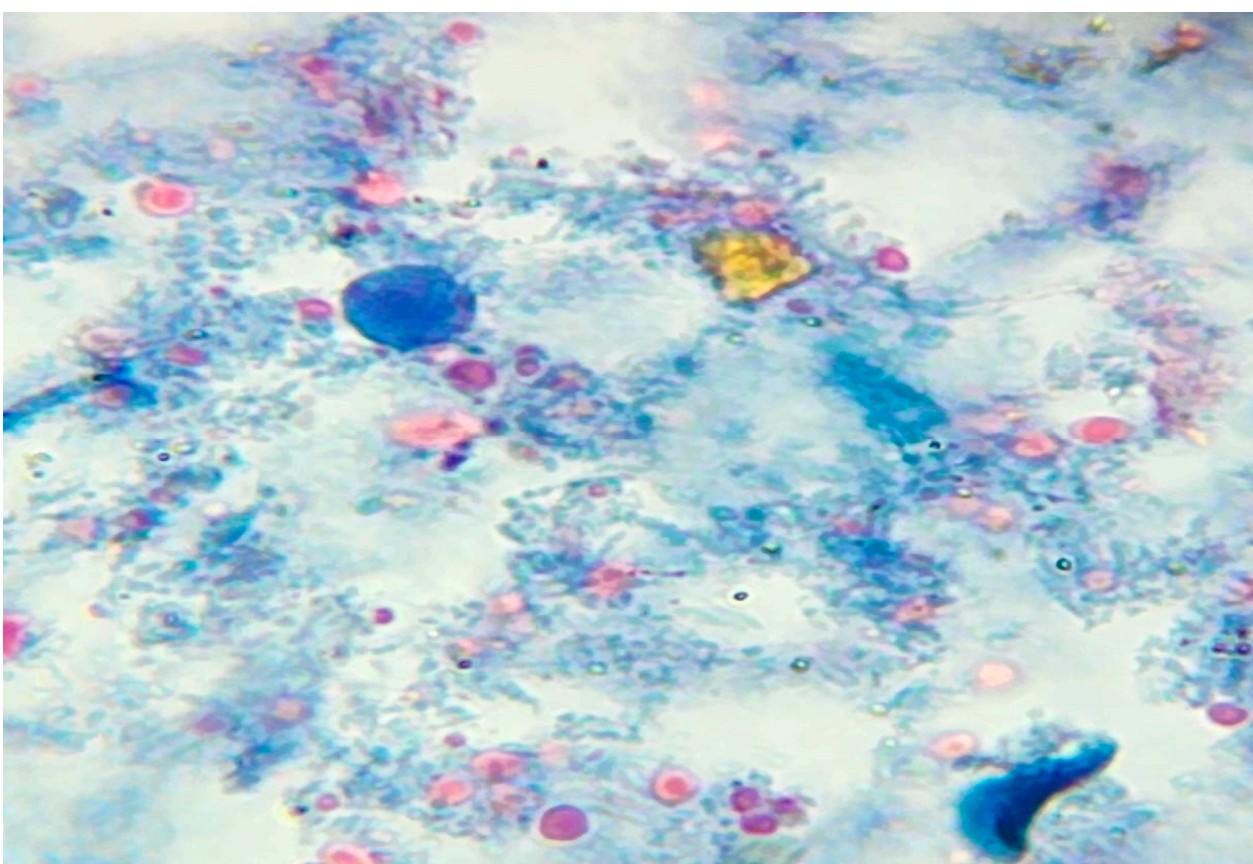

**Figure 2.** Vacuolar form of *B. hominis* stained with trichrome stain.

In conclusion, our study showed that patients with autoimmune illnesses had a prevalence rate of *Blastocystis* infection of 11.5%. Our results provide support for the idea that *B. hominis* infections may play a part in the development of autoimmune disorders. Future prospective cohort studies will be required to demonstrate that a *Blastocystis* infection worsens an already-existing autoimmune condition. It was also proposed that future studies might focus on subtyping the clinical isolates of *Blastocystis*. If such a study is conducted, it could be possible to determine whether there is a common *B. hominis* clone that is more common in those who have autoimmune illnesses. Until then, we advise checking the study population—particularly those with autoimmune diseases—for the presence of *Blastocystis*. Hematological anomalies could be caused by *Blastocystis hominis*. Another healthy population cohort study is advised for establishing the cause-and-effect connection. Future research on this subject is obviously necessary to clarify how this parasite interacts with the host's immunological inflammatory response; however, it is possible that *Blastocystis* and some particular subtypes could provide an anti-inflammatory situation.

## 5. Conclusions

Our results are consistent with the theory that *B. hominis* infection may contribute to the pathogenesis of autoimmune diseases in this study's context. This possibility necessitates further prospective cohort studies to establish which serotype is associated with autoimmune diseases and suggests that study populations should undergo *B. hominis* screening to prevent autoimmune diseases from worsening.

**Author Contributions:** Conceptualization, K.A.I.; resources, M.M.A.-w.; writing—original draft preparation, K.A.I., O.M.K. and A.M.M.; writing—review and editing, M.M.A.-w. and H.M.H.; statistical analysis: A.M.M.; supervision, K.A.I. and M.K.M. All authors have read and agreed to the published version of the manuscript.

**Funding:** This research received no external funding.

**Institutional Review Board Statement:** This study was approved by Taif University's and KFMC ethical council as being ethical.(H-02-T-123).

**Informed Consent Statement:** All patients who were recruited for the study were informed of its goals and its methods.

**Data Availability Statement:** Data available on required.

**Conflicts of Interest:** The authors declare no conflict of interest.

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
