# Peer review of "Molecular Identification of Blastocystis hominis Isolates in Patients with Autoimmune Diseases"

_2673-8007, doi:10.3390/applmicrobiol3020029_

Round 1

Reviewer 1 Report

The article presents the results of stool examination for the presence of Blastocystis in patients with autoimmune diseases. In my opinion, the article provides important information and further development of the research results in the future may contribute to the development of drugs or vaccines. However, in my assessment, the article is written in a somewhat chaotic manner, with incomplete and disjointed sentences, significantly lowering its value. Therefore, I would ask the authors to proofread the article, carefully read and complete the incomplete sentences. Additionally, I would like to request that section 2.1 titled "patients and settings" be reworded, as it is currently unclear. Furthermore, in section 2.3, did the authors use microscopic examination as a parasitological examination? If so, please clarify this.

English language correction is necessary.

Author Response

We would like to thank the reviewer for this valuable input and we have modified the manuscript as much as we can to make it more convenient in terms of the language. 

  • In section 2.1 we have changed the paragraph for more clarification.
  • In section 2.3 we explained that we did the microscopic examination (Wet mount and staining technique) as a parasitological examination.

Reviewer 2 Report

The authors researched the participation of Blastocystis hominis in gastrointestinal symptoms of patients with autoimmune diseases. They investigated 72 cases with gastrointestinal symptoms and 58 cases as controls. Their study revealed higher rate of infection in patients with systemic lupus erythematosus and ulcerative colitis than in controls, and better sensitivity by PCR assay than by wet amount microscopy. This manuscript is well written. However, several issues need to be addressed by authors.

Major comments:

1. They tested Blastocystis infection by microscopy and PCR assay. Therefore, they should describe the definition of “positive” and “negative” of Blastocystis infection of the Table 1 and 3 in the method section.

2. Patients with autoimmune diseases attending a hospital generally get some medication, such as immunosuppressants. They should describe the stool materials obtained before, under or after medication. If they were got under or after medication, the authors had also better show the types of medication.

3. The authors described the distribution of Blastocystis among patients with each autoimmune disease. They should also describe the data of total patients they studied to prove their conclusion, “our study showed that patients with autoimmune illnesses had a higher prevalence rate of Blastocystis infection”, with scientific evidences.

Minor comments:

1. The third sentence in the paragraph of “2.1. Patients and settings:” (at the line 158-159) was difficult to follow.

All points should be included in the manuscript.

Author Response

We would like to thank the reviewer for this valuable feedback and here is our reply point by point:

Major comments:

1- In both tables 1,3 we added in their description that the positivity and the negativity depend on PCR detection which has high sensitivity and specificity.

2- All of these stool samples were obtained during the treatment of different autoimmune diseases and we mentioned this in the manuscript.

3- We have modified the conclusion to be consistent with our results.

Minor comments:

1- The sentence has been changed for more clarification.